# Analysis of Wind-Induced Vibrations on HVTL Conductors Using Wireless Sensors

**DOI:** 10.3390/s22218165

**Published:** 2022-10-25

**Authors:** Federico Zanelli, Marco Mauri, Francesco Castelli-Dezza, Davide Tarsitano, Alessandra Manenti, Giorgio Diana

**Affiliations:** Department of Mechanical Engineering, Politecnico di Milano, 20156 Milan, Italy

**Keywords:** transmission lines, monitoring, fymax, wireless sensors, accelerations, wind, vortex shedding, galloping, condition-based maintenance, fatigue

## Abstract

In a world accelerating the energy transition towards renewable sources, high voltage transmission lines represent strategic infrastructure for power delivery. Being slender and low-damped structures, HVTL conductors are affected by wind-induced vibrations that can lead to severe fatigue issues in conductors and other components. Vibration monitoring could represent a key activity to assess the safety level of the line and perform condition-based maintenance activities. This work proposes an innovative approach based on the knowledge of the physical phenomena and smart technological devices. A wireless monitoring system based on MEMS accelerometers and energy harvesting techniques has been designed to measure the fy_max_ parameter in the field, which represents a fatigue indicator useful to identify the different wind-induced phenomena and assess the conductors’ strain level. A field test on a Canadian transmission line was used in the check of the efficiency of the system and collection of significant data. Vibrations due to vortex shedding were identified with a maximum value of fy_max_ = 50 m/s, while subspan oscillation and galloping were not observed. We show the novel method can detect the different wind-induced phenomena and pave the way to the development of suitable software able to compute a conductor’s residual fatigue life.

## 1. Introduction

High voltage transmission lines (HVTL) are mechanical and civil structures built with the purpose of transferring electrical energy from the point where it is generated to the point where it is employed in different ways. These constructions cover distances of several hundreds of kilometers and, being very long-lasting assets, they require huge long-term investments to operate with reliability in time under different operative conditions [1].

A rising trend in electricity demand has been recently observed, and it is foreseen to continue in future years because electricity supply is becoming more and more fundamental to the needs of our society. Overhead lines still represent the most cost-effective solution compared with underground cables to transfer electrical energy; however, new lines are nowadays more difficult to build due to several constraints including public opinion, environmental issues, and economics. Therefore, the most adopted solution in countries endowed of a widespread electrical network, like Europe, is to uprate the capacity of already existing transmission lines [2].

Under this framework, many existing transmission lines are approaching their service life both from a technical and economical perspective; and for this reason, companies managing these infrastructures have begun to develop maintenance strategies with the goal of extending the operational life of a line based on its condition. Therefore, many lines built in the past are now going to play a more significant role, and consequently their reliability must be improved.

In structural terms, HVTLs are essentially composed of towers and conductors, which account for about 30–35% of total line investments [1]. Conductors especially represent the most important component in transmission lines since their purpose is to transfer the electrical energy, and their cost with respect to the total investment of the line is very relevant, as already pointed out.

From a structural point of view, wind action on HVTL represents the most important action for the design of towers and fittings. Many failures of towers have been recorded due to high wind [3]. Moreover, in cold regions, the wind action can combine with the presence of ice on cables resulting in even higher loads on the structure. However, the static load acting on the towers is not the only problem to be solved in the design of HVTL. Conductors, in fact, are stranded cables endowed of very low structural damping, and this makes them very susceptible to wind induced oscillations generated by the fluid-structure interaction. The wind action on conductors produces three main wind-induced phenomena in different conditions, which are aeolian vibrations, subspan oscillations, and galloping. Although many research works have been conducted in past years and can be found in the literature, subspan oscillations and galloping are not yet fully understood in their physical mechanism. Aeolian vibrations are instead more investigated, also because they represent the most occurring instability on HVTL conductors. A representation of the typical motion shape of the three different mechanisms is shown in Figure 1.

Aeolian vibrations are called those motions related to vortex-induced vibrations (VIV), caused by the occurrence of vortex shedding from the conductors [4,5]. This phenomenon is generally characterized by small amplitudes of vibrations and relatively high frequencies. Typical aeolian vibration frequencies lie between 4 and 120 Hz, and the maximum antinode amplitude does not exceed the conductor diameter. The frequency of vibration of forces generated by vortex shedding *f_st_* can be approximately obtained through the Strouhal formula (Equation (1)).
(1)$$f_{st}=St\frac{V}{D},$$
where *V* represents the wind speed, *D* is the conductor diameter and *St* is the Strouhal number, whose value can be considered equal to 0.185 for transmission line conductors [6]. If the level of vibrations produced by vortex shedding is not suitably mitigated, the cyclic bending stresses imposed by aeolian vibrations represent the main cause of conductor fatigue failures. Due to its high occurrence, in fact, this phenomenon can lead to fatigue issues of conductors and of the other components installed on them, such as dampers, spacers, warning spheres, etc. In the case the damping level is increased up to a sufficient value, the vibration amplitude tends to zero, although Equation (1) remains satisfied. Equation (2), which represents the expression of the Scruton number, a non-dimensional parameter identifying VIV severity, explains this concept
(2)$$S_{c}=2\pi \frac{hm}{\rho D^{2}}.$$
where *h* is the non-dimensional damping, *m* is the conductor mass per unit length, *D* is the conductor diameter and *ρ* is the air density.

Therefore, to mitigate the aeolian vibrations phenomenon, it is important to increase the damping of the system through suitable devices, such as Stockbridge dampers and spacer-dampers (for bundle conductors).

Bundle conductors can be affected also by subspan oscillations, which are excited by the wake effect experienced by sub-conductors when the wind is blowing on the bundle [7,8]. The phenomenon takes place for medium-to-high wind speed (8–10 m/s ≤V≤ 15–20 m/s) and it is characterized by high amplitudes of vibration (reaching values equal to the bundle separation) and low frequency of vibration (0.7–2.5 Hz). The study of the phenomenon by means of 2DOF relying on the quasi steady theory (QST) to model the aerodynamic forces revealed that subspan oscillations are an instability mechanism caused by the non-conservative field of forces generated on the leeward cylinder by the wake of the windward one [9,10,11]. This phenomenon is therefore a flutter type of instability characterized by two different frequencies along the horizontal and vertical directions, which become equal due to the fact that the positional components of the aerodynamic forces are able to introduce energy in the system [9]. The peculiarities of this mechanism make it very difficult to predict such dynamic behavior by means of a mathematical model, and another cause of complexity derives from the fact that many parameters can influence the sensitivity of a line to this phenomenon.

Subspan oscillations can cause the spacer clamp loosening in case it is poorly designed or inefficient, leading to conductor strand failures due to the abrasion and hammering between the clamp body and the conductor.

In the end, galloping is an instability phenomenon caused by the formation of ice accretion on the conductors, which produce an unstable shape [12,13,14]. This condition gives rise to very large amplitude motions that can lead to failure of both conductor fittings and tower components, and to contacts among different phases. The galloping excitation mechanism can mainly be of two different types. The first is the so-called “Den Hartog instability”, which is a one degree-of-freedom aerodynamic instability caused by the fact that, due to the ice shape present on the conductor, the lift and drag coefficients provide the negative aerodynamic damping (Equation (3)) [15],
(3)$$\frac{\partial C_{L}}{\partial \alpha }+C_{D}<0.$$
where *C_L_* is the lift coefficient, *C_D_* is the drag coefficient and *α* is the wind angle of attack. The second form of excitation mechanism is flutter aeroelastic instability that can occur, in particular, on bundled conductors. This kind of mechanism is a two degree-of-freedom instability usually involving the vertical and torsional motions [14,16,17]. The reason for which bundled conductors are particularly susceptible to this phenomenon is that their vertical, horizontal and torsional modes natural frequencies can be very close to each other. Usually, it is the coupling between vertical and torsional motion that leads to the extreme galloping phenomena in bundled conductors.

Because galloping is a very complex phenomenon, it is still not completely defined in all its aspects; performing a prediction on the dynamic behavior of a galloping conductor by means of a mathematical model can be a very difficult task [18,19,20].

The consequences of unsuitable mitigated ice galloping are represented by short circuits due to the contact between different phases that lead to conductor strand burn and breakage and, in worst cases, failures of insulator strings and tower members. The most dramatic observed effect is represented by tower collapses, which can trigger a cascade failure of several sections of a transmission line.

Clearly, then, it is of the utmost importance to continuously monitor the occurrence of these phenomena in time and the possible onset of issues on the line caused by them.

However, a real time monitoring system to detect the wind-induced phenomena acting on the conductors of a transmission line and measure the induced level of strains is not currently available on the market.

Vibration measurements in the field are often conducted on conductors after the construction of a new line to verify the efficiency of the design and install a damping system. In this case, a couple of spans with the greatest exposure to the wind are usually considered for vibration monitoring activities as representative of the whole line. Spans surrounded by open flat terrain, where a significant transversal wind is expected; crossing spans on rivers, valleys and fiords; and spans located in critical zones with ice formation can all represent good choices for this aim.

However, the growing trend in the field is to also perform vibration measurements on existing lines, in some peculiar spans to assess the structural obsolescence of both conductor and fittings, and pursue a condition based maintenance (CBM) approach for the different components eventually present in a span. Typical situations are represented by cases in which failures due to fatigue or corrosion have been found on the line, the conductor type is critical in relation to the environmental and pollution conditions, or some parameters (such as span lengths and T/w) are well above standard values. Moreover, for aged lines, as a result of modelling, a possible critical state of solicitation could be found in some sections of the line, because the efficiency of the installed damping system has decreased over time or environmental actions have increased with respect to climate change considerations in the design stage [21]. In this context, severe events like downbursts are occurring more frequently and can have a serious impact on transmission lines, which are not designed to withstand these phenomena [22,23].

Vibration measurements have been performed in the past with several different devices, ranging from vibration detectors that were only able to provide an approximate indication of the vibration level, to generic transducers such as accelerometers, strain gauge anemometers, etc., usually employed in outdoor test stations but sometimes also used on operative lines [24]. Moreover, optical devices, such as opto-electronic recorders and laser recorders, have also been adopted for this purpose and were able to measure very small vibration amplitudes. They can detect possible fatigue issues occurring on conductors and fittings; however, they are not intended to be installed for a large amount of time.

To evaluate the vibration severity on the conductor, measurements were firstly performed through the use of strain gauges at the mouth of the suspension clamp, which should represent the most critical point for fatigue issues [25,26]. Through this method, the computation of the remaining conductor fatigue life is straightforward since the measured data can be directly associated with the fatigue curves of the conductor, commonly expressed as strain/stress versus number of cycles to failure.

Nevertheless, measuring bending strains at the clamp mouth did not represent a feasible solution for field applications. For this reason, the so-called “bending amplitude method” was proposed in 1964 [27] to adopt a more practical and measurable parameter directly linked to the bending strain at the mouth of the suspension clamp.

This method, in fact, suggests the measurement of the bending amplitude Yb, which is defined as the peak-to-peak displacement of the conductor relative to the suspension clamp, at 89 mm from the LPC between the clamp and the conductor (Figure 2). The reason for the chosen distance can be found in the fact that in this zone the conductor vibration is affected only by the stiffness effect and not influenced by inertia forces.

To obtain an idealized bending stress in the outer layer strands at the mouth of the suspension clamp, Poffenberger and Swart studied the dynamic deflection field of the conductor near a fixed clamp and formulated Equation (4), which is commonly known as the “Poffenberger-Swart formula” [28],
(4)$$\sigma_{a}(Y_{b})=\frac{E_{a}dp^{2}}{4(e^{-px}-1+px)}Y_{b}.$$
where *Y_b_* is the peak-to-peak bending amplitude, *E_a_* is the Young modulus of the outer-layer wire material, *d* is the diameter of outer layer wire, *p* is a parameter equal to $\sqrt{\frac{T}{EI_{min}}}$, *T* is the conductor tensile load during the monitoring period, *EI_min_* is the sum of flexural stiffnesses of wires in the cable while *x* represents the distance between the measurement point and LPC.

Equation (4) is therefore useful to compute the idealized bending stress in the top-most outer-layer strands of the conductor in the LPC.

Then, taking advantage of the theories developed by Poffenberger and Swart and Palmgren-Miner, Cigrè WG 22-04 provided an approximate method to compute the remaining fatigue life of a conductor under the effect of aeolian vibrations. The main recommendations valid for the vibration monitoring of aeolian vibrations can be found in [29].

Concerning the devices adopted for measuring the bending amplitude in the field, suitable vibration recorders have been developed and are commercially known as “bending amplitude recorders”. An example of one of these instruments installed on the line is shown in Figure 3.

As can be appreciated in Figure 3, bending amplitude recorders are installed at the suspension clamp and usually equipped with an LVDT displacement transducer. Inverted bending amplitude recorders are instead mounted on the conductor, where the device measures the motion directly above the LPC between the clamp and conductor.

The acquisition procedures are usually performed by these devices for 10 s every 15 min, since the framework of the bending amplitude method suggests conducting measurements at regular intervals in time to capture the structural response under different environmental conditions. After the signal acquisition and elaboration, data are stored in matrices containing a number of frequency and amplitude classes. In each matrix cell, therefore, the number of events recorded for each class of amplitude/frequency combination is stored.

Aside from vibration data, temperature and wind speed measurements are also performed and saved in separate arrays, if the devices are equipped with the appropriate sensors. Concerning wind data, however, the information stored regards the number of times the wind speed overpassed a certain value over the number of total events; therefore, it is not possible to directly correlate the single vibration event with the wind features occurring at that time.

Even if the bending amplitude method was developed considering metal-to-metal suspension clamps, bending amplitude recorders are nowadays used to monitor vibrations on spans equipped with elastomer-lined clamps and clamps with helical rod attachments, regardless of this assumption.

From the hardware point of view, bending amplitude recorders are battery powered and their autonomy depends on the sampling parameters and environmental conditions, which have an impact on the battery ageing [30]. The autonomy available for measurements can be estimated between a few weeks to some months and, as a result, these devices are typically employed in field tests with a one-month duration.

This timeframe, however, might not be in line with the minimum test period required for an effective monitoring activity. A suitable time window can be identified in the period necessary to cover the whole range of speeds able to excite the frequency range of interest of aeolian vibrations. This value can be identified in months, but it has to be remarked that in most cases wind and terrain conditions change seasonally, and therefore measurements must also be performed under the most severe environmental conditions. The measurement reliability of this method, in fact, is based on the concept that data are acquired considering few spans for a limited amount of time, and this information should be representative of the predominant condition that the entire line will face during its expected service life. Therefore, the correct choice of the test location, duration and season is very critical for the reliability of bending amplitude measurements.

It is then evident that a device permanently installed on some peculiar spans can represent the best solutions in this framework. Devices constantly mounted on the structure could in addition avoid issues in the instrumentation mounting and dismantling from the conductors, an operation that usually requires the outage of operative lines.

In the end, bending amplitude measurements may be affected by some inaccuracies due to the device attachment to the suspension clamp. The main inaccuracies sources are represented by an erroneous measurement of the recorder lever arm or a mistaken sensor rest position. Noteworthy, several papers report cases of measurements influenced by the not negligible mass and moment of inertia of the recorder itself, which could play a role if the device is not correctly mounted as visible in Figure 3. This fact is of particular relevance for monitoring activities on small conductors. The natural frequency of the vibration recorder plus clamping system can also affect the measurement if its value is not sufficiently above the range of frequencies of aeolian vibrations.

In the end, real time information from the line is not available, meaning that the correct functioning of the device cannot be remotely assessed, nor a warning sent for a possible dangerous situation happening on the line to the TSO.

In conclusion, a new method is considered to detect, in a straightforward and efficient way, large amplitudes of vibration possibly occurring at midspan and the level of strains in the corresponding fittings (such as spacers, vibration dampers, warning spheres, etc.) where fatigue issues can also occur in addition to the suspension clamp section. The idea proposed in this study is to develop an innovative and smart wireless monitoring system able, through acceleration measurements, to measure in real time a fatigue indicator with the goal of detecting possible issues on the monitored line.

## 2. Method

As pointed out in the introduction, the bending amplitude method does not actually represent the best solution to perform on-the-field continuous vibration monitoring of HVTL conductors with the aims of identifying the different wind-induced phenomena and detecting the strain level to determine if the line is in safe condition.

The idea proposed in this paper is to obtain for the first time from the field through acceleration measurements a fatigue indicator already standardly employed in fatigue tests carried out in a laboratory span, which is represented by fy_max_ [31]. This parameter is the product of the antinode vibration amplitude y_max_ (which is the maximum amplitude of vibration of a free loop in the span) by the vibration frequency f (Figure 4).

In Figure 4, the free span vibration angle *β* is related to the wavelength *λ* and *y_max_* through Equation (5).
(5)$$\beta =\frac{2\pi }{\lambda }y_{\max}=2\pi fy_{\max}\sqrt{\frac{m}{T}},$$
the *fy*_max_ term represents a more practical parameter to be measured in the field with respect to *Y_b_*, and through its knowledge it could be possible to assess fatigue issues also in points far from the suspension clamps that could be critical for fatigue, such as near clamps of spacers and dampers, and near warning spheres.

As previously found [32], the measurements of relative amplitudes near span extremities provide only a part of the vibration information, which has to be integrated by in-span measurements to conduct a reliable vibration risk analysis. This fact becomes particularly relevant when in-span devices such as vibration dampers are mounted on the line, since in those points the conductor motion is restrained and may be at risk of fatigue failures.

Another key advantage is the fact that by measuring *fy*_max_ in real time it is straightforward to detect which kind of wind-induced mechanism is affecting the span during the life of the structure and understanding with statistical analysis if this parameter changes in time suggesting a degradation of the damping capacity of the line.

*fy*_max_ directly represents an index of the level of strain present on the conductor. As in the case of the bending amplitude *Y_b_*, the *fy*_max_ parameter can also be used to compute an idealized dynamic stress *σ_a_*, which represents a sort of “nominal” conductor stress on the outer layer, useful to assess the effect of a certain vibration level to the mechanical safety level of a conductor. Thus, the relation between *fy*_max_ and *σ_a_* expressed by Equation (6) is only valid to compute bending stresses at the suspension clamp if the conductor is not equipped with dampers [6].
(6)$$\sigma_{a}(fy_{\max})=\pi dE_{a}\sqrt{\frac{m}{El}}fy_{\max},$$
where *E_a_* is the Young’s modulus of the outer layer wire material, *d* is the diameter of the outer layer wire, *f* the vibration frequency, *y_max_* represents the antinode vibration amplitude, *m* is the conductor mass per unit length and *EI* is the sum of flexural stiffnesses of individual wires in the cable.

The innovative idea at the basis of the method proposed in this study is then to measure the maximum amplitudes of vibration occurring on a single subspan inside the monitored span and correlate them with stresses in the various section of interest through numerical simulations performed with suitable software. This way, the number of sensor nodes to be employed for the monitoring activity is limited and the configuration of the real span can be reproduced in every significant feature into the numerical code.

*fy*_max_ measurements are also useful to get knowledge of the amount of stress excited on the conductor in the case of galloping phenomena and subspan oscillations [33]. As previously intimated, in these cases, fatigue issues are less likely to occur since those phenomena rarely happen with respect to aeolian vibrations, and cumulated cycles may not be sufficient to produce damage on the conductor. Nonetheless, they can both lead to severe damage of conductors and fittings as explained in Section 1.

From this analysis, it is therefore evident how could be significant to be able to perform continuous monitoring of this fatigue indicator on operative lines to assess the strain level and consequently safe working conditions. This could be achieved using a suitable monitoring system composed of a limited number of sensors.

In the case of aeolian vibrations, in fact, the spatial turbulence of the wind very often produces the simultaneous excitation of two or more slightly different vibration frequencies in separate locations along the span. This behavior can be caused by the variation of the wind speed mean value, especially in the case of long crossings. Beats are then produced by the overlapping of these vibrations and, as a result, a continuous variation of the vibration amplitude in any point of the span can be observed. An example of beats coming from the experimental data collected in the field test of Section 4 is shown in Figure 5. The three time-histories, referring to different positions along the span, have been reconstructed using vibration data collected by sensor nodes and show vibration characterized by very similar frequencies but different in amplitude. This behavior is due to the fact that aeolian vibrations in a bundle are characterized by vibration amplitudes that remain constant between two consecutive spacers but change from one subspan to the next, as reported in [6]. This situation occurs for the modes typical of the bundle in which the spacers’ features play a significant role. Studies showed that in reality the aeolian vibration phenomenon is not characterized by the presence of nodes or antinodes; therefore, only one measurement point is required to obtain the maximum level of vibration present in the span by means of the developed monitoring system. The proposed method aims in this context to conduct measurements of the fy_max_ parameter at regular intervals in time with continuity in specific points of the span, and numerically obtain mode shapes and strains by developing suitable software. An example of sensor positioning along a span of quad bundle conductors is visible in Figure 6, where two sensor nodes, in a crisscrossed configuration (one on one of the upper subconductors and one on one of the lower subconductors) in the first subspan, can be used to detect both aeolian vibrations and galloping phenomena, while two sensors one on each of the upper subconductors in one of the longest subspans close to midspan can be used for the identification of subspan oscillation, in addition to the other two aforementioned phenomena.

Through this method, it is thus possible to employ a limited quantity of sensors, suitably positioned inside the span chosen for the monitoring activity, to detect aeolian vibrations, subspan oscillations and galloping phenomena.

## 3. Description of the Developed Monitoring System

A suitable monitoring system was designed to measure and monitor the fy_max_ parameter with continuity in time. A full description of the developed monitoring system, together with some laboratory tests carried out to check its efficiency, is reported in detail in [34,35].

The monitoring system developed for the vibration monitoring of HVTL conductors is overall composed by a certain number of wireless sensor nodes and a gateway, whose role is to receive and store all data coming from the nodes and send them remotely using the GSM connection. The sensor nodes must be mounted on the conductors to measure the vibration level, while the gateway can be installed on the tower nearest to the instrumented span. The communication range between the sensor nodes and gateway is in fact in the order of hundreds of meters, as shown in [34,35]. A representation of the proposed architecture is visible in Figure 7.

The wireless sensor nodes can transmit the acquired data to the acquisition mini pc through the master board, which serves the role of managing the messages exchange between the sensor nodes and acquisition PC. All data coming from the active sensor nodes are therefore captured wirelessly by the master board and transmitted via serial communication to the PC, where they are pre-processed and stored. The main features of the acquisition PC are a very low power consumption (approximately 10 W) and ability to automatically power on in case of electrical outages. A block diagram that illustrates the devices composing the gateway is shown in Figure 8.

An essential device of the gateway is represented by the weather station, since it collects wind data in terms of wind speed, wind direction with respect to the line and turbulence intensity. This parameter is hugely important since the wind is not constant due to the turbulence effect. The first effect of turbulence is that, as already pointed out, many vibration modes at a time are excited. The second one is that, depending on the type of terrain characterizing the surrounding of the line, turbulence intensity has an impact on the occurrence and severity of wind-induced vibrations by affecting the wind power input. Namely, it has been found from experimental tests that as turbulence increases, the amplitude of vibration decreases (i.e., the energy introduced by the wind decreases) [36,37].

In addition to wind data, it is possible to collect other useful environmental information, such as temperature, humidity, dew point, etc. This kind of data can also be directly correlated to vibration data.

From the electrical point of view, the whole gateway can be fed by a 220 V power supply when available in the monitoring area; otherwise, it can be power supplied by batteries recharged when needed by means of energy harvesters (i.e., PV panels and/or mini wind turbines), as visible in Figure 9 [38].

Regarding wireless sensor nodes, they have been designed to perform accelerometric measurements on HVTL conductors subject to wind-induced vibrations for measuring the fy_max_ parameter. For this reason, they have been named “WindNodes”. Nonetheless, thanks to the inherent versatility, their range of application could be extended to any mechanical or civil structure affected by vibrations by suitably modifying their firmware according to the specific application [39,40].

To perform continuous monitoring activities, WindNodes have been developed to be energetically autonomous through the balance between very low mean power consumption and energy inflow coming from an energy harvester. The energy harvester adopted for this device is a mini photovoltaic (PV) panel, which can obtain good performance in different monitoring activities [39,40].

The low energy consumption has been achieved through the selection of suitable electronic components and adoption of a smart state machine, which keeps the nodes in sleep mode for most of the time when they are not performing an acquisition task.

From the structural point of view, sensor nodes have been designed to be compact and light devices to not have an influence on conductor vibrations, and consequently the measurements performed. The 3D printed enclosure has been specifically designed to suitably contain all the electronic components. The realized electronic board is internally restrained to the enclosure through four screws, one in each corner. The influence of this solution on the measurement performance of the sensor has been deeply studied by means of shaker tests, with the aim of verifying that no resonance condition of the board is present in the frequency range of interest. The material chosen for the enclosure is chlorinated polyethilene (CPE), which is a plastic material very robust against shocks and environmental agents suitable for use in outdoor environments characterized by severe stresses.

To not influence the vibration mode shapes and thus the measurement, the sensor plus clamping system has been developed with a mass target below 1 kg. This value has been defined considering the mass per unit length of a standard conductor, which can be estimated to be approximately 2 kg/m. If the conductor is particularly small with a mass per unit length below 1 kg/m (as in the case of ground wires), the sensor including the clamp must have a weight below the one corresponding to a 1 m-length conductor. Overall, the weight optimization is mainly affected by the clamp shape, which is influenced by the conductor diameter, while the sensor part contribution is limited to approximately 0.2 kg.

The communication protocol chosen for the WindNode is the Bluetooth Low Energy (BLE). This protocol offers an excellent trade-off between long communication range and energy consumption [41]. The acquisition task is conducted by the on-board mounted microcontroller and MEMS accelerometer, which allow one to provide reliable measurements together with low power consumption and very compact dimensions. Data acquired are then processed onboard by performing the fast fourier transform (FFT) and synthetic data regarding harmonic oscillations with the greatest amplitudes are then transferred to the gateway.

To summarize, the main features of the developed WindNode are listed in Table 1.

The acquisition logic designed to perform useful measurements through the developed monitoring system finds its roots in the deep knowledge of the physical behavior of the phenomena under observation. As will be better explained in the following, in fact, the whole acquisition logic is driven by wind action and excited vibration frequencies.

Overall, the acquisition logic has been implemented in the monitoring system by means of the low-level firmware installed in sensor nodes and through the acquisition code running at high level in the gateway.

Starting from the low-level side, the firmware developed for sensor nodes is subdivided into two main parts, one running in the sensor node microprocessor, whose role is to perform the computational work, and the other on the BLE transceiver, whose task is to manage the communication side.

The WindNode operativity is characterized by an optimized duty cycle, consisting of a working period, which corresponds to the data acquisition and transmission stages, followed by a sleep one. During the sleep time, which can be increased or decreased according to the monitoring requirements through the acquisition software, the microprocessor goes into an ultra-low-power mode, while the communication module is put in a reset state. Under these conditions, sensor node power consumption is very low. In standard conditions, the sleep time is set to 30 s, since this value allows to obtain periodic measurements that, in the case of a slowly variable phenomenon as the wind, can be seen as a truly continuous monitoring activity. Meanwhile, the chosen sleep time value grants a long autonomy, enhanced by the use of solar energy harvesting.

The communication between the sensor nodes and gateway is realized by means of messages exchange between the two entities. The acquisition process begins when WindNodes wake up and send this information to the gateway. Sensor nodes can perform the acquisition in two different modes, depending on the phenomena to be detected. This definitely represents the core of the whole logic.

As pointed out in Section 1, the different wind-induced mechanisms occur in specific range of the wind speed and are characterized by different vibration frequencies. Therefore, we created a standard “Aeolian” acquisition mode to detect aeolian vibrations, which are the most recurrent phenomenon observed in HVTL, and one “Subspan/Galloping” mode to detect with more accuracy very low frequency phenomena, as in the case of subspan oscillations and galloping.

The selection between the mode to be performed is taken on the basis of the mean wind speed measured in the field. As it can be appreciated in Figure 10, in fact, subspan oscillation and galloping occur more frequently in the presence of medium-to-high wind speed, while aeolian vibrations are more prone to be excited by wind speeds below 7 m/s.

This value is therefore taken as the threshold for the selection of the acquisition mode to be run by the gateway, as can be seen in Figure 11, where the high-level state machine is schematized.

Under standard conditions, the gateway sends to the sensor nodes a start message to perform acquisitions of the vibrations in “Aeolian” mode (Mode A), which is characterized by the sampling of 512 points at 400 Hz. The sampling of 512 points is the maximum permitted by the hardware adopted, while the 400 Hz sampling frequency obtains a time window and frequency resolution suitable to investigate aeolian vibrations. The FFT is performed in this case over a limited frequency range (i.e., 100 Hz) to avoid high frequency noise. This frequency range is in accordance with what is experimentally observed for vortex shedding-related phenomena, which usually occurs in the approximate range 3–100 Hz [42]. When the threshold on the mean wind speed is overpassed, the PC automatically triggers sensor nodes to the “Subspan/Galloping” mode (Mode B), characterized by a longer acquisition time (i.e., approximately 20 s) and lower sampling frequency (i.e., 25 Hz). A low pass filter is also switched on in the sensor nodes microcontroller. In this way, a higher frequency resolution, useful to detect the very low frequency phenomena with more accuracy, is obtained. Sensor nodes remain in this modality until the mean wind speed measured on the field drops under the threshold or after 10 consecutive acquisition cycles (N_cycle), when they automatically return to the standard “Aeolian” mode. Other conditions could be added and implemented to detect subspan oscillations and galloping, such as the occurrences of vibrations characterized by very low frequency and high amplitudes or the presence of a low environmental temperature suggesting the presence of an ice layer on the conductors.

Independently from the mode selected for the acquisition, the microcontroller of each node acquires the signals from the Y and Z axes of the accelerometer (lateral and vertical acceleration, respectively, see Figure 12) and computes data by performing the FFT.

The gateway looks for the maximum amplitude detected and related frequency among all data received from WindNodes and asks each sensor to send the spectrum line corresponding to that frequency (as well as the two adjacent lines). This operation is performed to get the information about the most excited mode, which is the one that mainly contributes to fatigue issues on the conductor. Moreover, taking advantage of the knowledge of the spectrum lines adjacent to the central one (spaced approximately 1 Hz from i), it is possible to reconstruct with accuracy the time history of the beats that contain the excited modes. In addition, through the installation of more sensor nodes in suitable positions on the conductors, this would give the possibility of reconstructing that mode shape.

Meanwhile, the PC acquires wind data from the weather station, in terms of wind mean speed, and wind direction with respect to the line and turbulence intensity, correlating them to the vibration data just collected.

In the end, after the last acknowledge message, sensor nodes are put in sleep mode for the chosen amount of time and once this time has elapsed another acquisition cycle begins.

Besides vibration data, diagnostic information on the battery status (state of charge and voltage) and the detected temperature are included in the messages sent by sensor nodes to the gateway.

All data collected during the acquisition cycles are stored in the gateway and, through the use of a post-processing algorithm developed for the scope and running on-board, only significant data are uploaded in the cloud through the 4G connection to reduce the amount of information to be remotely transmitted.

## 4. Field Test and Experimental Results

The wireless monitoring system described in Section 3 was adopted for a monitoring campaign on a HTVL in Manitoba (Winnipeg, MB, Canada). The main purpose of this field test was the effectiveness check of the damping system installed on a 500 kV DC line. Recently, DC lines are becoming a popular choice to cover very long distances since they represent an advantage in economic terms [43]. This activity represented the opportunity to test for the first time the developed system in a real scenario in the quite challenging weather conditions typical of the Canadian winter.

The measurements were performed in a location characterized by a flat terrain with some woods in the neighborhood (Figure 13a). The monitored span was 485 m long and equipped with nine spacer-dampers. In the portion of the line under study, the conductor employed was an AAAC in a triple bundle configuration (Figure 13b).

Due to extreme environmental conditions expected in the field test, the most crucial components of the system were selected to withstand an extended temperature range. The accelerometer is characterized by an operating temperature range of −40–85 °C. This feature, together with the fact that the accelerometer is internally conditioned and gives as output a digital signal to the microcontroller, ensures the measurement reliability in the range of temperature of interest. Another crucial component to be carefully examined due to the temperature influence on its performance was the battery equipping sensor nodes. To this aim, a long campaign of laboratory tests was carried out to check the suitability of the chosen battery to operate in cold weather conditions [35].

Since no power supply was available on the site, the gateway was fed through two high performance batteries (with extended operative temperature range −40–45 °C) and recharged when needed by a hybrid energy harvesting system composed of a PV panel and mini wind turbine. The last element included in the experimental setup was a weather station necessary to conduct measurements on the wind speed and direction.

Through the design of a suitable clamping system, the gateway was mounted on the tower nearest to the monitored span, as shown in Figure 9. The weather station was placed at the standard height of approximately 10 m from obstacles that would have affected wind measurements, while the PV panel and wind turbine positioning was chosen to maximize their efficiency on the basis of the line orientation and direction of prevalent wind.

A suitable clamping system was designed to easily mount the sensor nodes on the conductors. The CPE enclosure of the sensor was inserted in the housing realized on the clamp (Figure 14), and use of a high-performance epoxy adhesive guaranteed a perfectly rigid connection. Moreover, in the design of the clamp particular attention was devoted to weight saving, with the aim of not influencing the measurements (as explained in Section 3), and to reduce the presence of edges in the component to avoid issues with the Corona effect (Figure 14) [44].

Sensor node positioning was instead chosen to be able to detect all three vibration phenomena, as explained in Section 2. Three sensor nodes were placed on the lower conductor in the second subspan to measure vibrations caused by vortex shedding, namely aeolian vibrations (Figure 15a). The other two devices were instead mounted in the longest subspan (inside the monitored span) on the two upper conductors to also detect the potential presence of subspan oscillations (Figure 15b). Both positions were also suitable for galloping measurements.

Throughout the three-month field campaign, many experimental data were collected. Especially worth highlighting is that the developed monitoring system provides the possibility to obtain results already in the frequency domain, with a direct correlation between vibration and wind measurements. Moreover, the deployment of the PV panel on the sensor nodes makes it suitable for long-term monitoring campaigns, as explained in [34,35].

Concerning vibration data, as already stated, sensor node acquisition outputs are in the form of maximum acceleration values detected and the corresponding frequency of vibration. The maximum vibration amplitudes are then obtained in Equation (7):(7)$$MaxAmp=\frac{MaxAcc\cdot g}{\omega^{2}},$$
where *MaxAcc* is the maximum acceleration detected by the sensor nodes expressed in [*g*], *g* is the acceleration gravity and *ω* is the circular frequency (*ω* = 2πf).

Firstly, it is interesting to analyze the relationship between the excited vibration frequency and wind speed acting on the line. As can be appreciated in Figure 16, the developed system provides insight into the lock-in effect, which characterizes the aeolian vibration phenomenon. The figure shows measurements performed over a time period of approximately 15 min, during which the vortex shedding frequency is locked-in to the cable natural frequency of 7.12 Hz. The vortex shedding is locked to the cylinder natural frequency in all the wind speeds, called the synchronization range, which is a very low Scruton number between 0.9 and 1.5 V/V_S_ (where V is the flow velocity and V_S_ the Strouhal velocity) [45]. These values are in good agreement with the data collected by the sensors, proving the possibility given by the developed method to deeply investigate aeolian vibrations on cables.

The developed system provides a deep understanding of the relationship between the wind acting on the line and resulting conductor’s vibration amplitude. An example is given by experimental data shown in Figure 17. In this three-dimensional graph, the maximum vertical vibration amplitudes are plotted against the wind speed and direction measured by the weather station.

Regarding wind direction, values around 0° and ±180° indicate a flow perfectly orthogonal to the line, while values around ±90° represent a wind flowing nearly parallel with respect to the considered span.

It can then be observed that the highest maximum vibration amplitudes are reached for wind speed up to 2 m/s and wind directions ranging between 40–50° and around −130°, which are values representing wind flows characterized by an important transversal component with respect to the line.

All these are typical hints of the presence of aeolian vibrations phenomenon.

It is then possible to analyze vibration data collected by focusing on the distribution of the maximum vibration amplitudes in the frequency domain. For this purpose, vibration data are divided into those coming from the “Aeolian” sensor nodes group and those from the “Subspan” one.

Figure 18a,b clearly shows that the higher vertical vibration amplitudes from the “Aeolian group” are reached for low frequencies, while horizontal vibration amplitudes show small values as one would have predicted in the case of aeolian vibrations. The maximum vertical vibration amplitude detected has a value of about 10 mm, which is not a negligible one, although this value occurs only once. Such an amplitude can be associated with a considerable solicitation, as will be shown in the following.

Switching to the “Subspan” group of nodes, it is evident from the measured amplitude values that no subspan oscillations occurred during the monitoring activity. In the frequency range of interest (1–2 Hz), in fact, no significant amplitudes either in the vertical or horizontal directions have been observed (Figure 19a,b). The values in the vertical direction can be reconducted again to vibrations due to vortex shedding but related to a different subspan.

In the end, the results are proposed in terms of fy_max_ which, as already stated in the introduction, is a standard way to express the strain experienced by the conductor, where f is the vibration frequency and y_max_ stands for the maximum antinode amplitude. This output is a significant result, since it provides an understanding of the actual stress level on the span directly from the measurements collected by the monitoring system. In Figure 20, the fy_max_ distribution is then plotted against the vibration frequency.

As can be noted, the maximum value corresponds to *fy*_max_ = 50 mm/s translating, by using Equation (8) for the classical computation of strains [6], into a value of approximately 100 μ*ε*.
(8)$$\varepsilon =K\cdot fy_{\max},$$
where *K* = 2.5 for the considered type of conductor.

This value is far from the fatigue limit of the conductor since, according to [6], the endurance limit of AAAC conductors is equal *fy*_max_ = 87 mm/s. Nonetheless, if this situation is repeated in time, it may suggest increasing the damping capacity of the instrumented span, although most values indicate that the actual stress level of the line is below the fatigue limit thanks to the energy dissipation provided by the installed spacer-dampers.

The experimental data collected thanks to the developed method are significant since this activity represents the first time in which information on the vibration level affecting the line is collected in real time and processed, gaining important knowledge on what is happening in the mechanical system composed of conductors and fittings.

## 5. Comparison between Experimental Data and Numerical Simulations

Experimental data collected during the field campaign were compared with results obtained following a numerical approach. The numerical simulations were performed through the use of ATTRA software developed at the Department of Mechanical Engineering of Politecnico di Milano, which is composed of a set of programs able to compute the dynamic behavior of conductors and conductor bundles subject to wind excitation. Bundles may be equipped either with spacers or spacer-dampers, taking into account the internal masses, and the stiffness and damping of the spacer; moreover, conductors and bundles may be equipped with dampers of known characteristics. ATTRA is based on the Energy Balance Principle (EBP) and computes vibrations based on both vortex shedding and subspan oscillations.

At first, the aeolian vibrations of bundle conductors program is employed to numerically reproduce the effect of vibrations induced by vortex shedding.

The first step is represented by the input of all data necessary to run the simulations. The modelling of the conductor is conducted by defining the stranding, and the main features of interest (conductor diameter, mass per unit length, reduced bending stiffness and ultimate tensile strength) are automatically computed by the program (Figure 21).

Boundary conditions are then imposed to reproduce the real configuration of the instrumented span, which is in this case a fully suspended one.

Regarding the span and environmental conditions data, it is important to note that a value of turbulence intensity equal to 15% is inserted because it was known from experimental wind data acquired on the field through the weather station. The role of this device in reproducing the correct level of turbulence inside numerical simulations is here highlighted.

Lastly, the features of the spacer-dampers installed on the line must be inserted as well as their positioning along the considered span, namely the staggering. This way, the number and length of the different subspans is defined. The modelled span and spacers staggering are visible in Figure 22.

After running the simulation, the numerical results in terms of maximum amplitudes of vibrations, fy_max_, and strains as a function of the vibration frequency are obtained.

In the following, the comparison between results obtained through numerical simulations and experimental data is conducted both in terms of maximum antinode amplitudes and fy_max_. The maximum amplitude of vibrations predicted by the numerical simulations are plotted in Figure 23a where red and purple curves refer, respectively, to maximum antinode amplitudes on the bundle conductors and maximum amplitude among all the spacers in the considered span. The blue curve represents the prediction for the case of the single conductor that is reported as a reference.

As can be observed in Figure 23a,b, experimental data exhibit a high coherence with respect to numerical results and accurately reproduce the distribution curve, especially in the medium-low frequency range (i.e., below 20 Hz) which can be considered the most significant in case of aeolian vibrations.

In the end, the subspan oscillation program is used for the remaining part of comparison. The input data are the same as the previous simulation, and it has been taken advantage of the possibility offered by the software to switch from one simulation mode to the other without the need to reintroduce the input data.

The first output of the subspan program is the computation of data of the instability field. In this framework, the program finds the combination of modes that can possibly couple to give rise to subspan oscillations. For each combination of modes, the wind speed range that can lead to instability is reported, together with the instability index, which is a measure of the possibility of instability occurrence.

In this case, however, numerical simulations ended with no graphical results because the bundle remains always stable for the considered wind speed range. This is consistent with the experimental data collected from the “Subspan” group of sensor nodes, since they were characterized by small vibration amplitudes not related to subspan oscillations.

In conclusion, the comparison conducted allowed us to assess the suitability of the developed monitoring system and reliability of the acquired data, which can be employed for the estimation of the stress level, as will be explained in future works. The next step is, in fact, represented by the development of a suitable software able to take advantage of the experimental data collected by the developed monitoring system to compute the conductor stress in every section of interest of the span. This way, it will be possible to compute the real stress level the line is subject to and obtain an estimation of the residual fatigue life of both conductors and fittings.

## 6. Conclusions

A novel method to perform vibration monitoring of transmission line conductors with the aim of detecting possible fatigue issues in some crucial sections of a span was illustrated in this study.

Starting from the analysis of the state of the art in the field and an in-depth study of the main wind-induced phenomena, a smart high-level algorithm, ruled by the wind speed acting on the line, has been created. An innovative monitoring system composed of a gateway and some sensor nodes has been designed and realized to obtain synthetic indices representative of the real level of vibration present in the conductors. The low-level firmware was written to make sensors able to work in two different acquisition modes to identify aeolian vibrations and the low-frequency phenomena (subspan oscillations and galloping) with more accuracy. The wireless sensors have been moreover developed considering the necessities of realizing energy autonomous and wireless devices, which are essential features to effectively perform continuous monitoring activities on HVTL.

The complete method was then adopted for the first time in a field test in Manitoba (Winnipeg, MB, Canada) on a triple bundle HVDC transmission line. This activity represented the first time in which vibration data were collected and analyzed in real time from an operating transmission line. The system was able to acquire significant data during the field campaign. A maximum value of fy_max_ = 50 mm/s was measured and typical examples of aeolian vibrations were detected, while neither subspan oscillations nor galloping phenomena were observed during the field test.

A numerical-experimental comparison conducted by means of well-established software showed very good agreement between the acquired data and simulation results. Through the development of suitable software, the proposed method can be used to compute the stress level in every point of the span and conduct an estimation of the residual life of both conductors and fittings cumulating stresses in time, leading to the possibility of performing condition-based maintenance activities in this field.

## Figures and Tables

**Figure 1 sensors-22-08165-f001:**
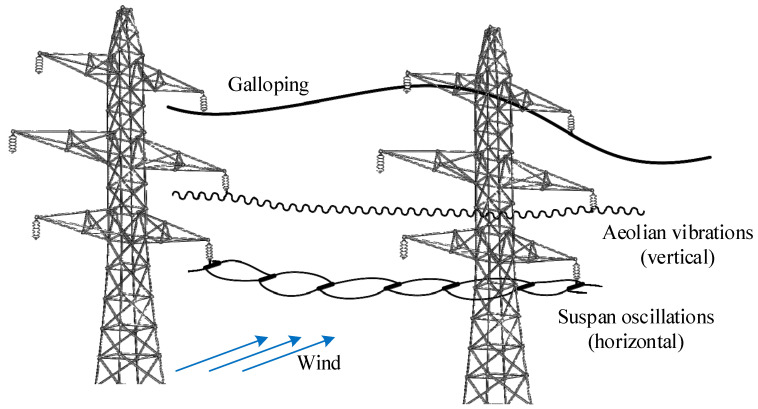
Typical motion shapes of the different wind-induced phenomena acting on HVTL conductors.

**Figure 2 sensors-22-08165-f002:**
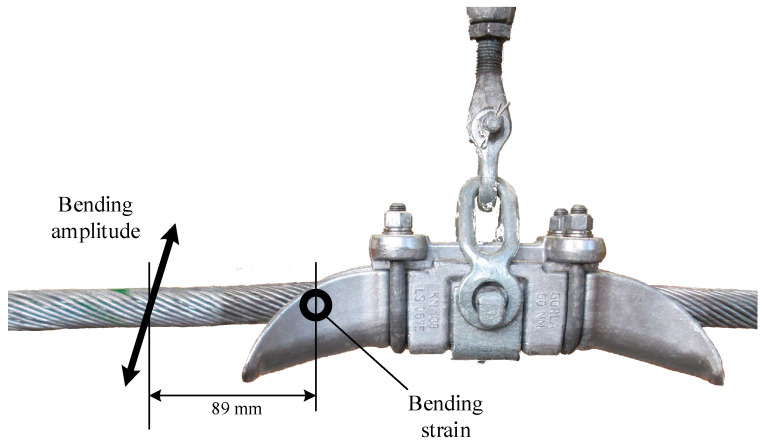
Representation of the main parameters at the basis of the bending amplitude method.

**Figure 3 sensors-22-08165-f003:**
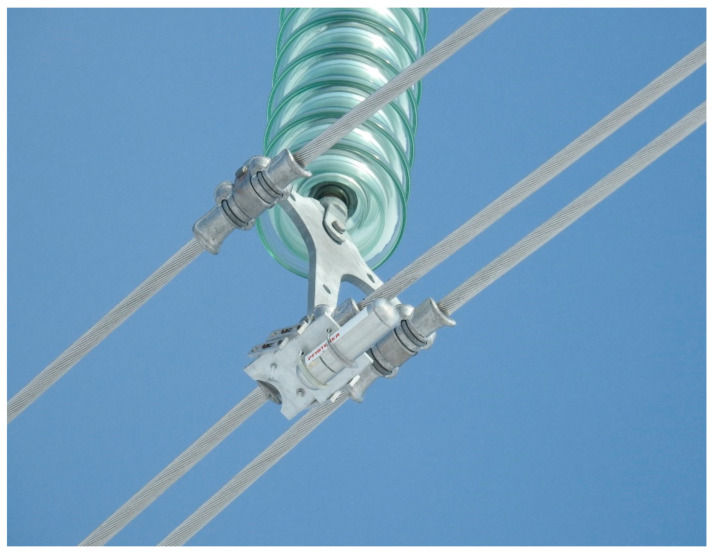
Vibration recorder mounted at the suspension clamp.

**Figure 4 sensors-22-08165-f004:**
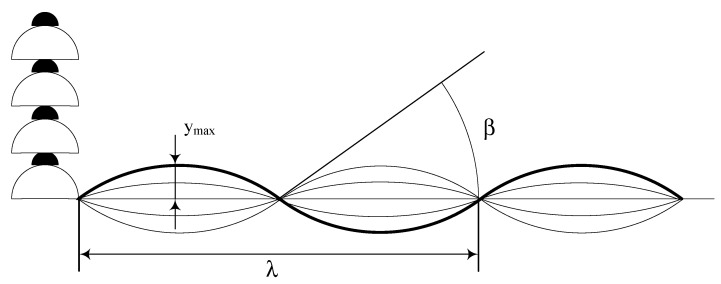
Schematization of some parameters useful for aeolian vibration studies.

**Figure 5 sensors-22-08165-f005:**
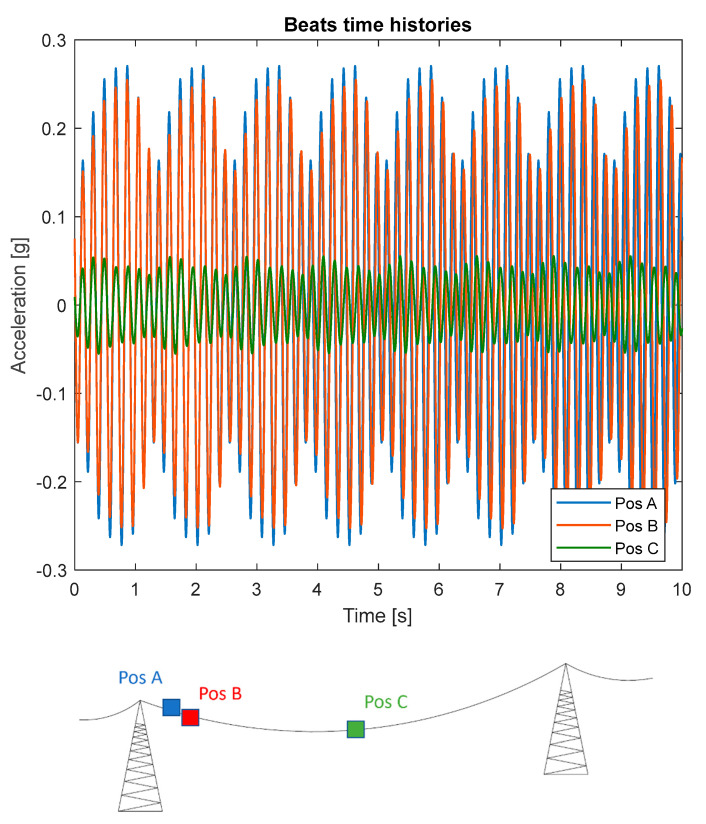
Experimental measurements of beats characterizing the aeolian vibrations phenomenon.

**Figure 6 sensors-22-08165-f006:**
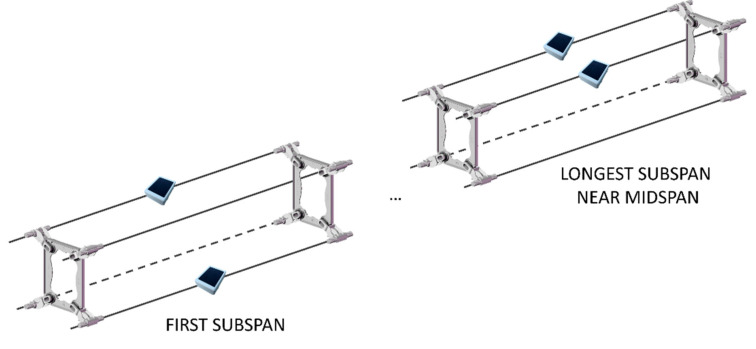
Schematization of possible sensor nodes positioning on a quad bundle.

**Figure 7 sensors-22-08165-f007:**
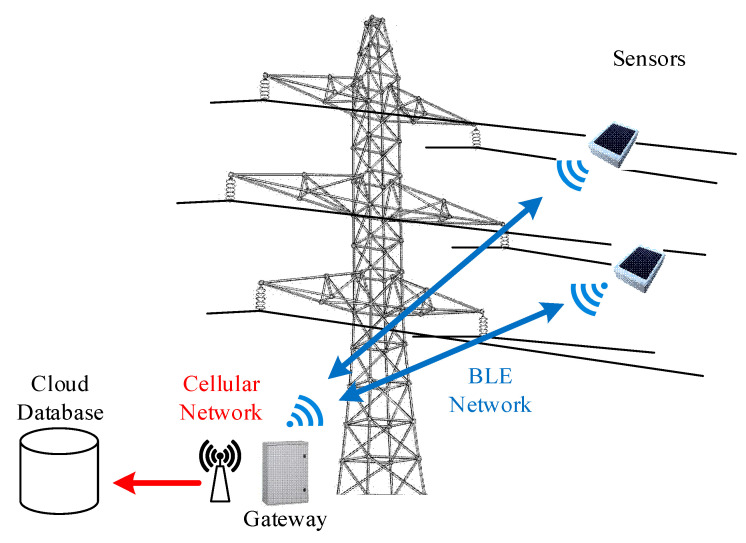
Architecture of the developed monitoring system.

**Figure 8 sensors-22-08165-f008:**
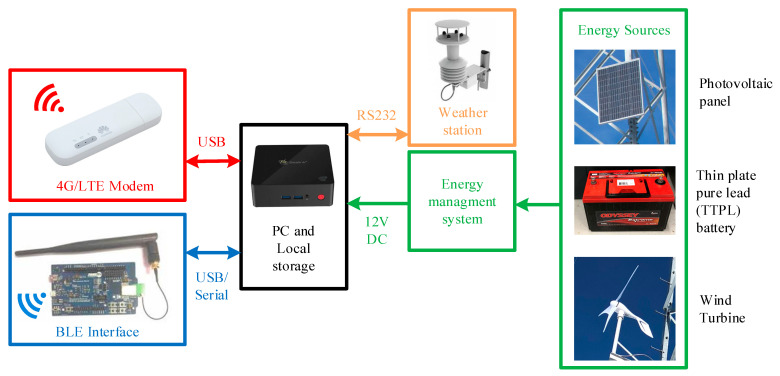
Block diagram of the devices composing the gateway.

**Figure 9 sensors-22-08165-f009:**
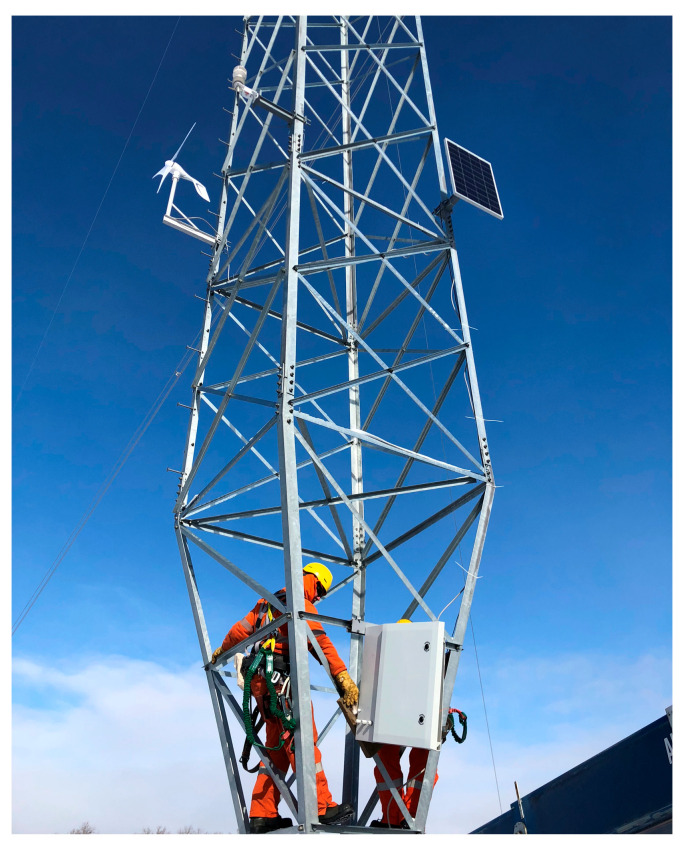
Gateway mounted on the tower nearest to the instrumented span.

**Figure 10 sensors-22-08165-f010:**
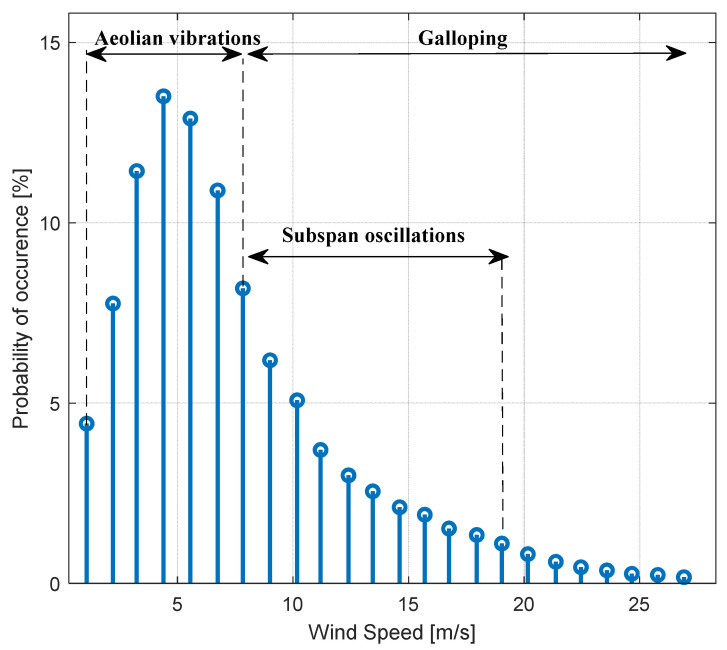
Probability of occurrence of the different mechanisms as a function of the wind speed.

**Figure 11 sensors-22-08165-f011:**
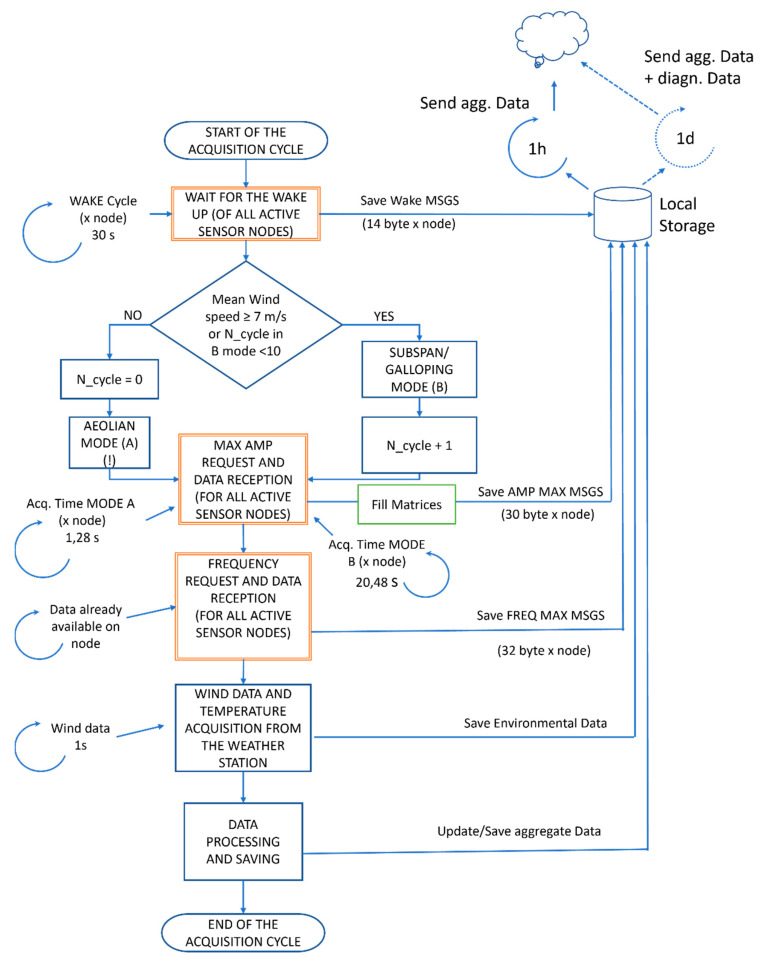
High-level algorithm flowchart.

**Figure 12 sensors-22-08165-f012:**
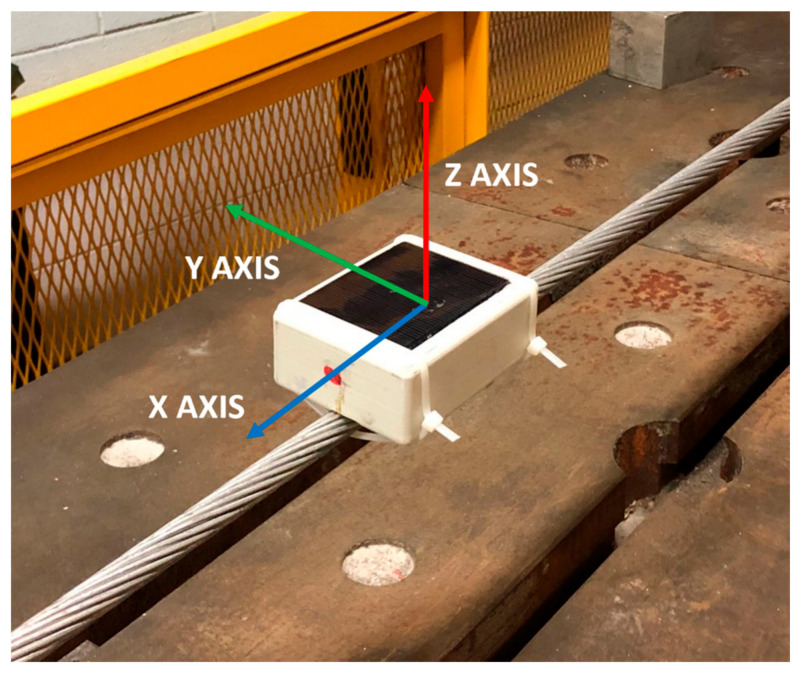
Accelerometer reference system with respect to the sensor node positioning on the conductor.

**Figure 13 sensors-22-08165-f013:**
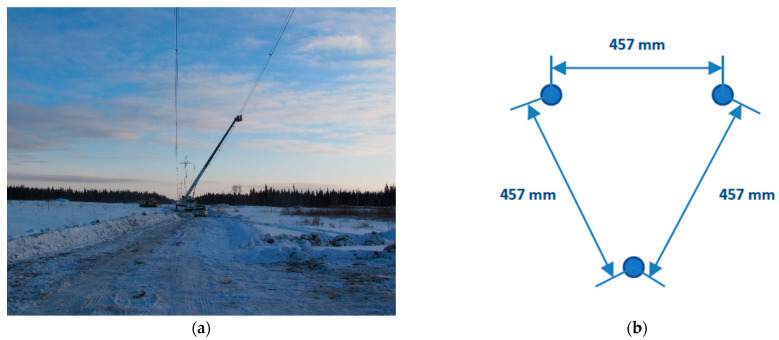
(**a**) Surroundings of the field test location; (**b**) Triple bundle configuration.

**Figure 14 sensors-22-08165-f014:**
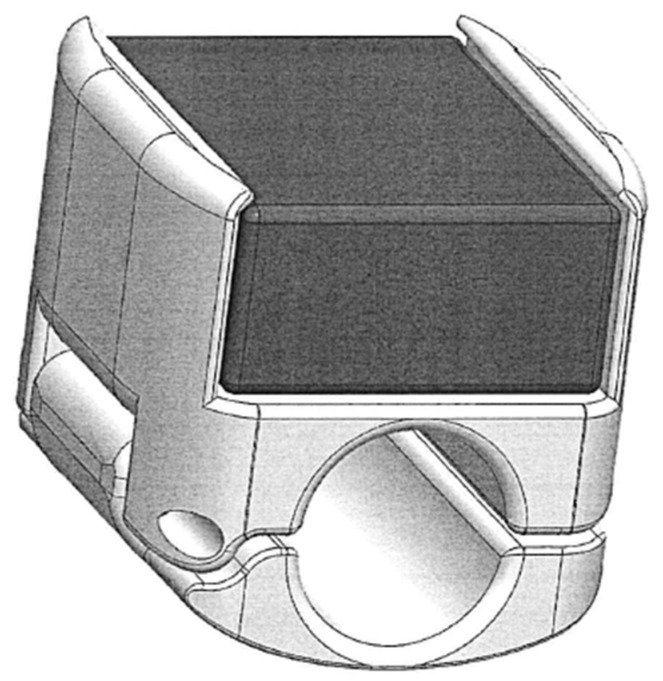
Rendering of the clamping system design for the installation of sensor nodes on the conductors.

**Figure 15 sensors-22-08165-f015:**
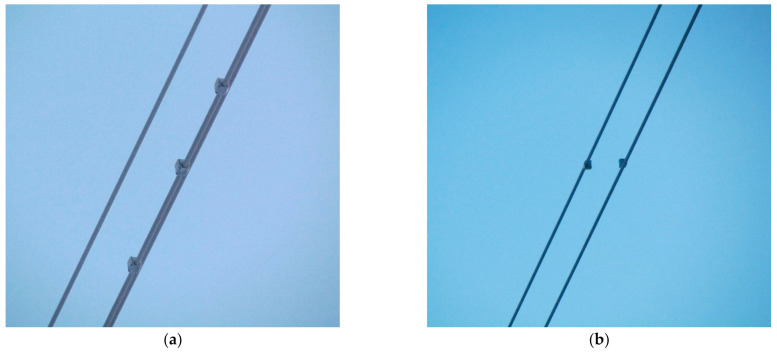
Sensor nodes positioning inside the span: (**a**) configuration for the detection of aeolian vibrations and galloping; (**b**) configuration to detect subspan oscillations.

**Figure 16 sensors-22-08165-f016:**
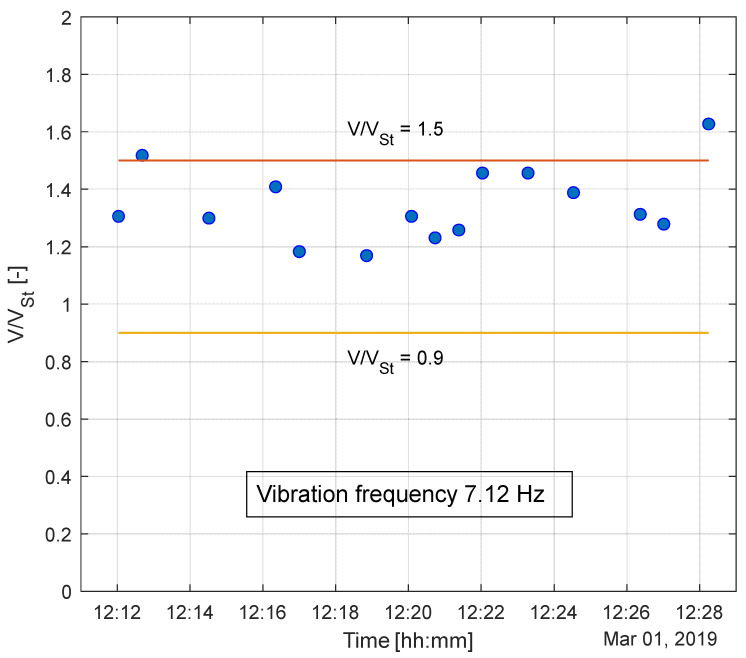
Experimental data referring to the lock-in condition characterizing the vortex shedding phenomenon with a focus on the synchronization range.

**Figure 17 sensors-22-08165-f017:**
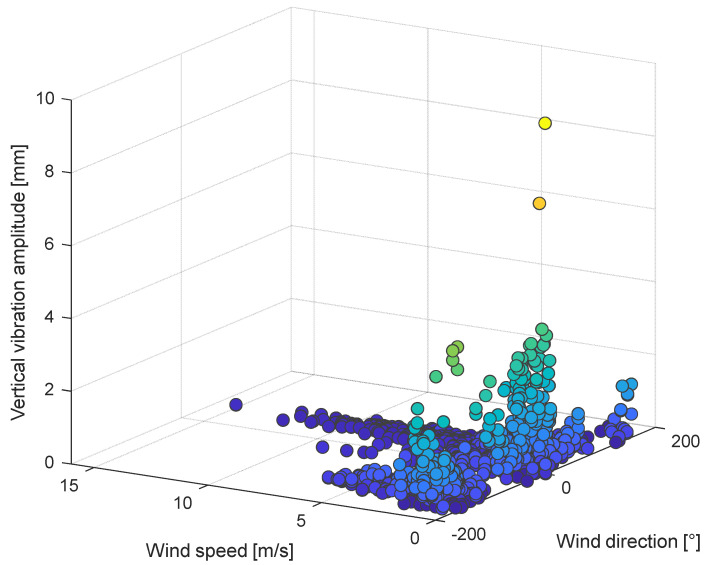
Correlation between maximum vertical amplitudes of vibration, wind speed and wind direction. Lighter colors indicate higher values of vibration amplitudes.

**Figure 18 sensors-22-08165-f018:**
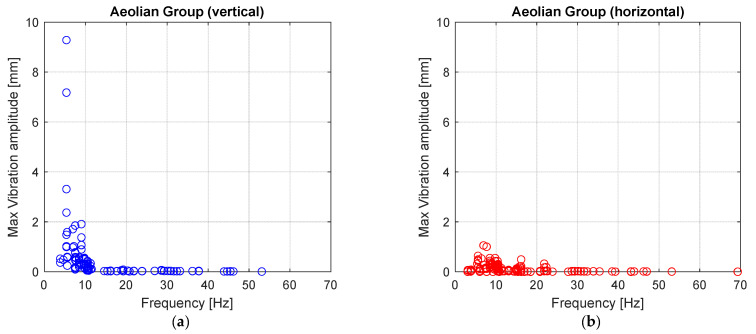
(**a**) Maximum vertical vibration amplitude and (**b**) maximum horizontal vibration amplitudes versus vibration frequency for “Aeolian” group.

**Figure 19 sensors-22-08165-f019:**
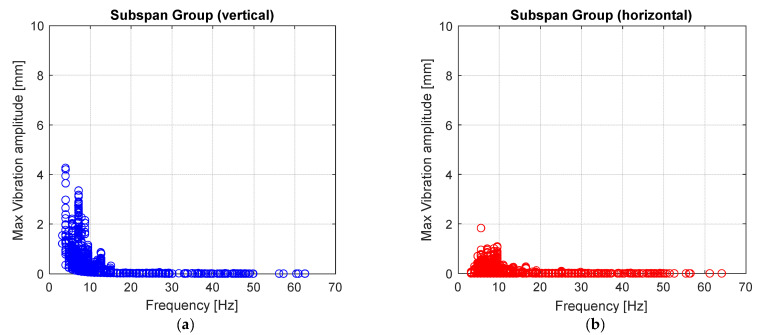
(**a**) Maximum vertical vibration amplitude and (**b**) maximum horizontal vibration amplitudes versus vibration frequency for “Subspan” group.

**Figure 20 sensors-22-08165-f020:**
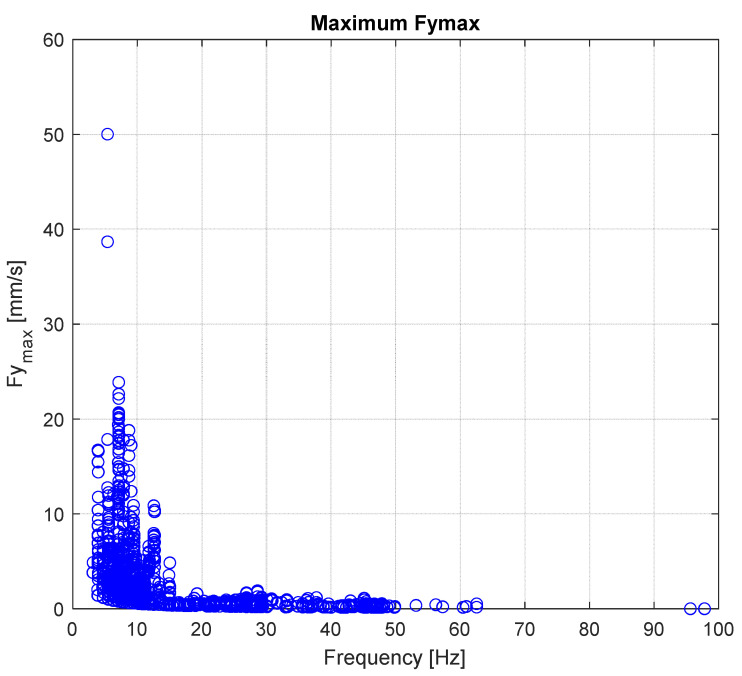
Maximum fy_max_ versus vibration frequency.

**Figure 21 sensors-22-08165-f021:**
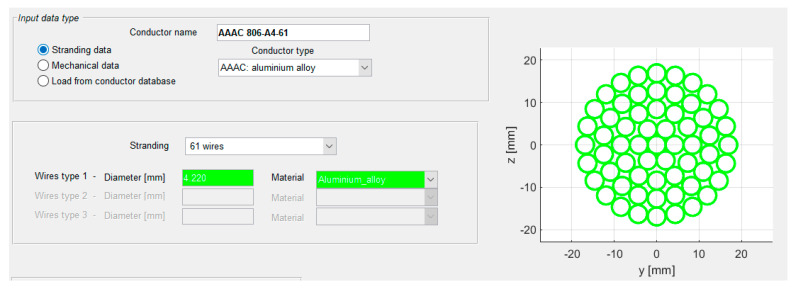
Conductor modelling in ATTRA software.

**Figure 22 sensors-22-08165-f022:**
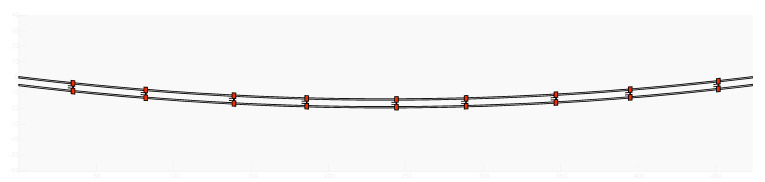
Schematization of the span modelled in ATTRA software.

**Figure 23 sensors-22-08165-f023:**
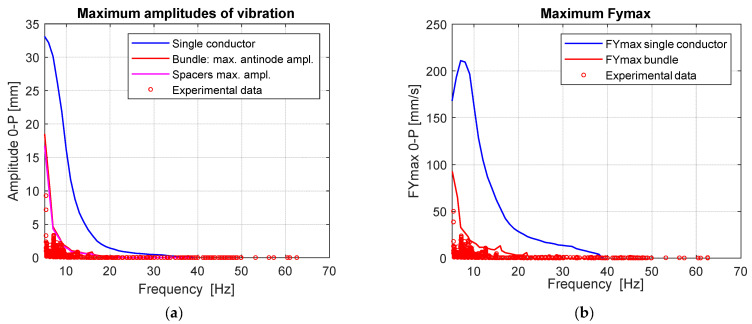
Comparison between experimental and numerical results: (**a**) Maximum antinode amplitudes as function of vibration frequency; (**b**) Maximum fy_max_ as function of vibration frequency.

**Table 1 sensors-22-08165-t001:** Main features of the WindNode prototypes.

Sensor Node Main Features	
Dimensions	91 × 70 × 38 mm
Weight	≈240 g
Communication protocol	Bluetooth Low Energy (BLE)
PV panel	5.5 V–0.5 W
Li-Po battery	3.7 V–2000 mAh
Enclosure material	CPE
Accelerometer model	Analog Devices ADXL 345
Accelerometer full-scale range	±16 g
Accelerometer sensitivity	256 LSB/g

## Data Availability

Not applicable.

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
