# Peer review of "Analysis of Wind-Induced Vibrations on HVTL Conductors Using Wireless Sensors"

_sensors, 2022, doi:10.3390/s22218165_

Round 1

Reviewer 1 Report

The paper is alright in the background. The authors of the paper present an interesting system that enables diagnostics of high-voltage overhead lines. The presented diagnostic system is based on two separate subsystems: wind velocity measurement system and a system based on accelerometers recording acceleration in two axes. In the presented diagnostic system, the operation of the algorithm (fig. 11)  managing the inference process is mainly based on the measurement of wind velocity. In the system, the frequency of sampling the measurement signals in the form of the acceleration of the overhead line's movement depends on the wind velocity.

Remarks:

- line 607 fig. 16 -  the graph is identical to that in the article "Design and Field Validation of a Low Power Wireless Sensor Node for Structural Health Monitoring .." p. 16 fig. 12

- line 445 table 1. - the table does not show the parameters of the accelerometric sensor.

- the article and the reference articles lack a description of attaching the accelerometer to the housing base. The method of mounting affects the reliability of the obtained measurements.

- line 493 - on what basis the time window of 512 samples and the sampling frequency was determined? How you set the range of natural frequencies (AEOLIAN VIBRATION)

- there is a systematic error in the work in the form of: FigureError! Reference source not found.  E.g. line 507

- please explain and present the influence of temperature on the reliability of the research. The presented system works in a wide range of temperatures.

Reviewer 2 Report

The authors have done some commendable work, however a few issues are left, which are needed to be addressed before proceeding further. 

1. The novel aspects of the work must be properly highlighted in the main manuscript. 

2. There are a lot of typo errors that needs to be addressed. 

3. The conclusion must be strongly written by highlighting core aspects of the findings. 

4. Kindly address the citation errors in the manuscript. 

5. What is figure X on page (14 line 481) and page (17 line 507)? Also where is equation X on page 20 line 28?  
